# FjORD: Fair and Accurate Federated Learning under heterogeneous targets with Ordered Dropout

**Samuel Horváth**[*]
KAUST[†]
Thuwal, KSA
samuel.horvath@kaust.edu.sa

**Stefanos Laskaridis**[*]
Samsung AI Center
Cambridge, UK
mail@stefanos.cc

**Mario Almeida**[*]
Samsung AI Center
Cambridge, UK
mario.a@samsung.com

**Ilias Leontiadis**
Samsung AI Center
Cambridge, UK
i.leontiadis@samsung.com

**Stylianos I. Venieris**
Samsung AI Center
Cambridge, UK
s.venieris@samsung.com

**Nicholas D. Lane**
Samsung AI Center &
University of Cambridge
Cambridge, UK
nic.lane@samsung.com

## Abstract

Federated Learning (FL) has been gaining significant traction across different ML tasks, ranging from vision to keyboard predictions. In large-scale deployments, client heterogeneity is a fact and constitutes a primary problem for fairness, training performance and accuracy. Although significant efforts have been made into tackling statistical data heterogeneity, the diversity in the processing capabilities and network bandwidth of clients, termed as system heterogeneity, has remained largely unexplored. Current solutions either disregard a large portion of available devices or set a uniform limit on the model's capacity, restricted by the least capable participants. In this work, we introduce Ordered Dropout, a mechanism that achieves an ordered, nested representation of knowledge in deep neural networks (DNNs) and enables the extraction of lower footprint submodels without the need of retraining. We further show that for linear maps our Ordered Dropout is equivalent to SVD. We employ this technique, along with a self-distillation methodology, in the realm of FL in a framework called FjORD. FjORD alleviates the problem of client system heterogeneity by tailoring the model width to the client's capabilities. Extensive evaluation on both CNNs and RNNs across diverse modalities shows that FjORD consistently leads to significant performance gains over state-of-the-art baselines, while maintaining its nested structure.

## 1 Introduction

Over the past few years, advances in deep learning have revolutionised the way we interact with everyday devices. Much of this success relies on the availability of large-scale training infrastructures and the collection of vast amounts of training data. However, users and providers are becoming increasingly aware of the privacy implications of this ever-increasing data collection, leading to the creation of various privacy-preserving initiatives by service providers [3] and government regulators [10].

Federated Learning (FL) [46] is a relatively new subfield of machine learning (ML) that allows the training of models without the data leaving the users' devices; instead, FL allows users to collaboratively train a model by moving the computation to them. At each round, participating devices download the latest model and compute an updated model using their local data. These locally trained models are then sent from the participating devices back to a central server where updates are

---

[*]Indicates equal contribution.
[†]Work while intern at Samsung AI Center.

35th Conference on Neural Information Processing Systems (NeurIPS 2021).

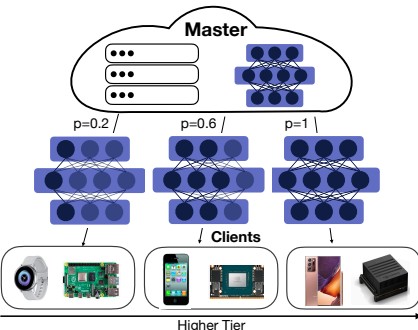

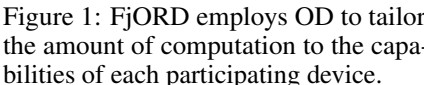

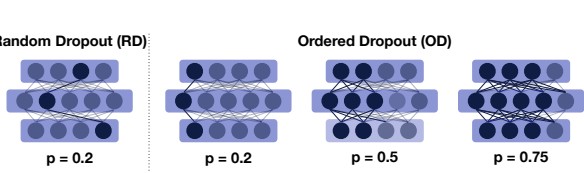

Figure 1: FjORD employs OD to tailor the amount of computation to the capabilities of each participating device.

Figure 2: Ordered vs. Random Dropout. In this example, the left-most features are used by more devices during training, creating a natural ordering to the importance of these features.

aggregated for next round's global model. Until now, a lot of research effort has been invested with the sole goal of maximising the accuracy of the global model [46, 42, 39, 31, 63], while complementary mechanisms have been proposed to ensure privacy and robustness [6, 14, 47, 48, 27, 4].

A key challenge of deploying FL in the wild is the vast *heterogeneity of devices* [38], ranging from low-end IoT to flagship mobile devices. Despite this fact, the widely accepted norm in FL is that the local models have to share the *same* architecture as the global model. Under this assumption, developers typically opt to either drop low-tier devices from training, hence introducing training bias due to unseen data [30], or limit the global model's size to accommodate the slowest clients, leading to degraded accuracy due to the restricted model capacity [8]. In addition to these limitations, variability in sample sizes, computation load and data transmission speeds further contribute to a very unbalanced training environment. Finally, the resulting model might not be as efficient as models specifically tailored to the capabilities of each device tier to meet the minimum processing-performance requirements [34].

In this work, we introduce FjORD (Fig. 1), a novel adaptive training framework that enables heterogeneous devices to participate in FL by dynamically adapting model size – and thus computation, memory and data exchange sizes – to the available client resources. To this end, we introduce Ordered Dropout (OD), a mechanism for run-time *ordered (importance-based) pruning*, which enables us to extract and train submodels in a nested manner. As such, OD enables *all* devices to participate in the FL process independently of their capabilities by training a submodel of the original DNN, while still contributing knowledge to the global model. Alongside OD, we propose a *self-distillation* method from the maximal supported submodel on a device to enhance the feature extraction of smaller submodels. Finally, our framework has the additional benefit of producing models that can be dynamically scaled during inference, based on the hardware and load constraints of the device.

Our evaluation shows that FjORD enables significant accuracy benefits over the baselines across diverse datasets and networks, while allowing for the extraction of *submodels* of varying FLOPs and sizes *without* the need for *retraining*.

## 2   Motivation

Despite the progress on the accuracy front, the unique deployment challenges of FL still set a limit to the attainable performance. FL is typically deployed on either siloed setups, such as among hospitals, or on mobile devices in the wild [7]. In this work, we focus on the latter setting. Hence, while cloud-based distributed training uses powerful high-end clients [19], in FL these are commonly substituted by resource-constrained and heterogeneous embedded devices.

In this respect, FL deployment is currently hindered by the vast *heterogeneity* of client hardware [66, 28, 7]. On the one hand, different mobile hardware leads to significantly varying processing speed [1], in turn leading to longer waits upon aggregation of updates (*i.e.* stragglers). At the same time, devices of mid and low tiers might not even be able to support larger models, *e.g.* the model does not fit in memory or processing is slow, and, thus, are either excluded or dropped upon timeouts from the training process, together with their unique data. More interestingly, the resource allocation to participating devices may also reflect on demographic and socio-economic information of owners, that makes the exclusion of such clients unfair [30] in terms of participation. Analogous to the device load and heterogeneity, a similar trend can be traced in the downstream (model) and upstream

(updates) network communication in FL, which can be an additional substantial bottleneck for the training procedure [55].

## 3 Ordered Dropout

In this paper, we firstly introduce the tools that act as enablers for heterogeneous federated training. Concretely, we have devised a mechanism of importance-based pruning for the easy extraction of subnetworks from the original, specially trained model, each with a different computational and memory footprint. We name this technique **Ordered Dropout** (OD), as it orders knowledge representation in *nested submodels* of the original network.

More specifically, our technique starts by sampling a value (denoted by $p$) from a distribution of candidate values. Each of these values corresponds to a specific submodel, which in turn gets translated to a specific computational and memory footprint (see Table 1b). Such sampled values and associations are depicted in Fig. 2. Contrary to conventional dropout (RD), our technique drops adjacent components of the model instead of random neurons, which translates to computational benefits[3] in today's linear algebra libraries and higher accuracy as shown later.

### 3.1 Ordered Dropout Mechanics

The proposed OD method is parametrised with respect to: i) the value of the dropout rate $p \in (0, 1]$ per layer, ii) the set of candidate values $\mathcal{P}$, such that $p \in \mathcal{P}$ and iii) the sampling method of $p$ over the set of candidate values, such that $p \sim D_{\mathcal{P}}$, where $D_{\mathcal{P}}$ is the distribution over $\mathcal{P}$.

A primary hyperparameter of OD is the dropout rate $p$ which defines how much of each layer is to be included, with the rest of the units dropped in a structured and ordered manner. The value of $p$ is selected by sampling from the dropout distribution $D_{\mathcal{P}}$ which is represented by a set of discrete values $\mathcal{P} = \{s_1, s_2, \ldots, s_{|\mathcal{P}|}\}$ such that $0 < s_1 < \ldots < s_{|\mathcal{P}|} \leq 1$ and probabilities $\mathbf{P}(p = s_i) > 0$, $\forall i \in [|\mathcal{P}|]$ such that $\sum_{i=1}^{|\mathcal{P}|} \mathbf{P}(p = s_i) = 1$. For instance, a uniform distribution over $\mathcal{P}$ is denoted by $p \sim \mathcal{U}_{\mathcal{P}}$ (*i.e.* $D = \mathcal{U}$). In our experiments we use uniform distribution over the set $\mathcal{P} = \{i/k\}_{i=1}^k$, which we refer to as $\mathcal{U}_k$ (or *uniform-k*). The discrete nature of the distribution stems from the innately discrete number of neurons or filters to be selected. The selection of set $\mathcal{P}$ is discussed in the next subsection.

The dropout rate $p$ can be constant across all layers or configured individually per layer $l$, leading to $p_l \sim D_{\mathcal{P}}^l$. As such an approach opens the search space dramatically, we refer the reader to NAS techniques [69] and continue with the same $p$ value across network layers for simplicity, without hurting the generality of our approach.

Given a $p$ value, a pruned $p$-subnetwork can be directly obtained as follows. For each[4] layer $l$ with width[5] $K_l$, the submodel for a given $p$ has all neurons/filters with index $\{0, 1, \ldots, \lceil p \cdot K_l \rceil - 1\}$ included and $\{\lceil p \cdot K_l \rceil, \ldots, K_l - 1\}$ pruned. Moreover, the unnecessary connections between pruned neurons/filters are also removed[6]. We denote a pruned $p$-subnetwork $\mathbf{F}_p$ with its weights $\boldsymbol{w}_p$, where $\mathbf{F}$ and $\boldsymbol{w}$ are the original network and weights, respectively. Importantly, contrary to existing pruning techniques [18, 35, 49], a $p$-subnetwork from OD can be directly obtained post-training without the need to fine-tune, thus eliminating the requirement to access any labelled data.

### 3.2 Training OD Formulation

We propose two ways to train an OD-enabled network: i) *plain OD* and ii) *knowledge distillation OD* training (OD w/ KD). In the first approach, in each step we first sample $p \sim D_{\mathcal{P}}$; then we perform the forward and backward pass using the $p$-reduced network $\mathbf{F}_p$; finally we update the submodel's weights using the selected optimiser. Since sampling a $p$-reduced network provides us significant computational savings on average, we can exploit this reduction to further boost accuracy. Therefore, in the second approach we exploit the nested structure of OD, *i.e.* $p_1 < p_2 \implies \mathbf{F}_{p_1} \subset \mathbf{F}_{p_2}$ and allow for the bigger capacity supermodel to teach the sampled $p$-reduced network at each iteration

---

[3]OD, through its nested pruning scheme that requires neither additional data structures for bookkeeping nor complex and costly data layout transformations, can capitalise directly over the existing and highly optimised dense matrix multiplication libraries.

[4]Note that $p$ affects the number of output channels/neurons and thus the number of input channels/neurons of the next layer. Furthermore, OD is not applied on the input and last layer to maintain the same dimensionality.

[5]*i.e.* neurons for fully-connected layers (linear and recurrent) and filters for convolutional layers. RNN cells can be seen as a set of linear feedforward layers with activation and composition functions.

[6]For BatchNorm, we maintain a separate set of statistics for every dropout rate $p$. This has only a marginal effect on the number of parameters and can be used in a privacy-preserving manner [41].

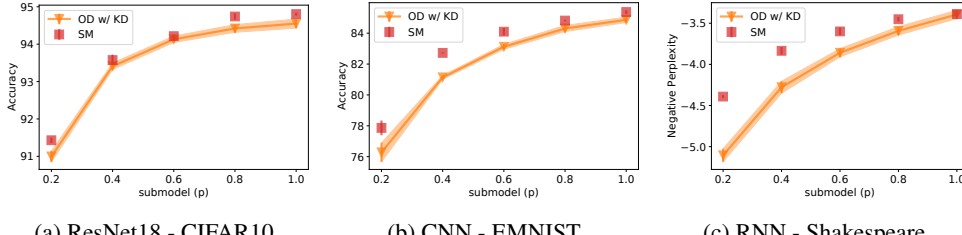

| (a) ResNet18 - CIFAR10 | (b) CNN - EMNIST | (c) RNN - Shakespeare |

Figure 3: Full non-federated datasets. OD-Ordered Dropout with $D_{\mathcal{P}} = \mathcal{U}_5$, SM-single independent models, KD-knowledge distillation.

via knowledge distillation (teacher $p_{\max} > p$, $p_{\max} = \max \mathcal{P}$). In particular, in each iteration, the loss function consists of two components as follows:

$$\mathcal{L}_d(\text{SM}_p, \text{SM}_{p_{\max}}, \boldsymbol{y}_{\text{label}}) = (1 - \alpha)\text{CE}(\max(\text{SM}_p), \boldsymbol{y}_{\text{label}}) + \alpha \text{KL}(\text{SM}_p, \text{SM}_{p_{\max}}, T)$$

where $\text{SM}_p$ is the *softmax* output of the sampled $p$-submodel, $\boldsymbol{y}_{\text{label}}$ is the ground-truth label, CE is the cross-entropy function, KL is the KL divergence, $T$ is the distillation temperature [21] and $\alpha$ is the relative weight of the two components. We observed in our experiments always backpropagating also the teacher network further boosts performance. Furthermore, the best performing values for distillation were $\alpha = T = 1$, thus smaller models exactly mimic the teacher output. This means that new knowledge propagates in submodels by proxy, *i.e.* by backpropagating on the teacher, leading to the following loss function:

$$\mathcal{L}_d(\text{SM}_p, \text{SM}_{p_{\max}}, \boldsymbol{y}_{\text{label}}) = \text{KL}(\text{SM}_p, \text{SM}_{p_{\max}}, T) + \text{CE}(\max(\text{SM}_{p_{\max}}), \boldsymbol{y}_{\text{label}})$$

### 3.3 Ordered Dropout exactly recovers SVD

We further show that our new OD formulation can recover the Singular Value Decomposition (SVD) in the case where there exists a linear mapping from features to responses. We formalise this claim in the following theorem.

**Theorem 1.** *Let $\mathbf{F}: \mathbb{R}^n \to \mathbb{R}^m$ be a NN with two fully-connected linear layers with no activation or biases and $K = \min\{m, n\}$ hidden neurons. Moreover, let data $\mathcal{X}$ come from a uniform distribution on the $n$-dimensional unit ball and $A$ be an $m \times n$ full rank matrix with $K$ distinct singular values. If response $y$ is linked to data $\mathcal{X}$ via a linear map: $x \to Ax$ and distribution $D_{\mathcal{P}}$ is such that for every $b \in [K]$ there exists $p \in \mathcal{P}$ for which $b = \lceil p \cdot K \rceil$, then for the optimal solution of*

$$\min_{\mathbf{F}} \mathbb{E}_{x \sim \mathcal{X}} \mathbb{E}_{p \sim D_{\mathcal{P}}} \|\mathbf{F}_p(x) - y\|^2$$

*it holds $\mathbf{F}_p(x) = A_b x$, where $A_b$ is the best $b$-rank approximation of $A$ and $b = \lceil p \cdot K \rceil$.*

Theorem 1 shows that our OD formulation exhibits not only intuitively, but also theoretically ordered importance representation. Proof of this claim is deferred to the Appendix.

### 3.4 Model-Device Association

**Computational and Memory Implications.** The primary objective of OD is to alleviate the excessive *computational* and *memory* demands of the training and inference deployments. When a layer is shrunk through OD, there is no need to perform the forward and backward passes or gradient updates on the pruned units. As a result, OD offers gains both in terms of FLOP count and model size. In particular, for every fully-connected and convolutional layer, the number of FLOPs and weight parameters is reduced by $K_1 \cdot K_2 / \lceil p \cdot K_1 \rceil \cdot \lceil p \cdot K_2 \rceil \sim 1/p^2$, where $K_1$ and $K_2$ correspond to the number of input and output neurons/channels, respectively. Accordingly, the bias terms are reduced by a factor of $K_2 / \lceil p \cdot K_2 \rceil \sim 1/p$. The normalisation, activation and pooling layers are compressed in terms of FLOPs and parameters similarly to the biases in fully-connected and convolutional layers. This is also evident in Table 1b. Finally, smaller model size also leads to reduced memory footprint for gradients and the optimiser's state vectors such as momentum. However, how are these submodels related to devices in the wild and how is this getting modelled?

**Ordered Dropout Rates Space.** Our primary objective with OD is to tackle *device heterogeneity*. Inherently, each device has certain capabilities and can run a specific number of model operations within a given time budget. Since each $p$ value defines a submodel of a given width, we can indirectly associate a $p_{\max}^i$ value with the $i$-th device capabilities, such as memory, processing throughput or energy budget. As such, each participating client is given at most the $p_{\max}^i$-submodel it can handle.

Devices in the wild, however, can have dramatically different capabilities; a fact further exacerbated by the co-existence of previous-generation devices. Modelling discretely each device becomes quickly

intractable at scale. Therefore, we cluster devices of similar capabilities together and subsequently associate a single $p_{\max}^i$ value with each cluster. This clustering can be done heuristically (*i.e.* based on the specifications of the device) or via benchmarking of the model on the actual device and is considered a system-design decision for our paper. As smartphones nowadays run a multitude of simultaneous tasks [43], our framework can further support modelling of transient device load by reducing its associated $p_{\max}^i$, which essentially brings the capabilities of the device to a lower tier at run time, thus bringing real-time adaptability to FjORD.

Concretely, the discrete candidate values of $\mathcal{P}$ depend on i) the number of clusters and corresponding device tiers, ii) the different load levels being modelled and iii) the size of the network itself, as *i.e.* for each tier $i$ there exists $p_{\max}^i$ beyond which the network cannot be resolved. In this paper, we treat the former two as invariants (assumed to be given by the service provider), but provide results across different number and distributions of clusters, models and datasets.

### 3.5 Preliminary Results

Here, we present some results to showcase the performance of OD in the *centralised* non-FL training setting (i.e. the server has access to all training data) across three tasks, explained in detail in § 5. Concretely, we run OD with distribution $D_{\mathcal{P}} = \mathcal{U}_5$ (uniform distribution over the set $\{i/5\}_{i=1}^5$) and compare it with end-to-end trained submodels (SM) trained in isolation for the given width of the model. Fig. 3 shows that across the three datasets, the best attained performance of OD along every width $p$ is very close to the performance of the baseline models. We extend this comparison against Random Dropout in the Appendix. We note at this point that the submodel baselines are trained from scratch, explicitly optimised to that given width with no possibility to jump across them, while our OD model was trained using a single training loop and offers the ability to switch between accuracy-computation points without the need to retrain.

## 4 FjORD

Building upon the shoulders of OD, we introduce FjORD, a framework for federated training over *heterogenous* clients. We subsequently describe the FjORD's workflow, further documented in Alg. 1.

As a starting point, the global model architecture, $\mathbf{F}$, is initialised with weights $\boldsymbol{w}^0$, either randomly or via a pretrained network. The dropout rates space $\mathcal{P}$ is selected along with distribution $D_{\mathcal{P}}$ with $|\mathcal{P}|$ discrete candidate values, with each $p$ corresponding to a subnetwork of the global model with varying FLOPs and parameters. Next, the devices to participate are clustered into $|\mathcal{C}_{\text{tiers}}|$ tiers and a $p_{\max}^c$ value is associated with each cluster $c$. The resulting $p_{\max}^c$ represents the maximum capacity of the network that devices in this cluster can handle without violating a latency or memory constraint.

At the beginning of each communication round $t$, the set of participating devices $\mathcal{S}_t$ is determined, which either consists of all available clients $\mathcal{A}_t$ or contains only a random subset of $\mathcal{A}_t$ based on the server's capacity. Next, the server broadcasts the current model to the set of clients $\mathcal{S}_t$ and each client $i$ receives $\boldsymbol{w}_{p_{\max}^i}$. On the client side, each client runs $E$ local iterations and at each local iteration $k$, the device $i$ samples $p_{(i,k)}$ from conditional distribution $D_{\mathcal{P}}|D_{\mathcal{P}} \leq p_{\max}^i$ which accounts for its limited capability. Subsequently, each client updates the respective weights $(\boldsymbol{w}_{p_{(i,k)}})$ of the local submodel using the `FedAvg` [46] update rule. In this step, other strategies [39, 63, 31] can be interchangeably employed. At the end of the local iterations, each device sends its update back to the server.

Finally, the server aggregates these communicated changes and updates the global model, to be distributed in the next global federated round to a different subset of devices. Heterogeneity of devices leads to heterogeneity in the model updates and, hence, we need to account for that in the global aggregation step. To this end, we utilise the following aggregation rule

$$\boldsymbol{w}_{s_j}^{t+1} \setminus \boldsymbol{w}_{s_{j-1}}^{t+1} = \text{WA}\left(\left\{\boldsymbol{w}_{i_{s_j}}^{(i,t,E)} \setminus \boldsymbol{w}_{s_{j-1}}^{(i,t,E)}\right\}_{i \in \mathcal{S}_t^j}\right) \tag{1}$$

where $\boldsymbol{w}_{s_j} \setminus \boldsymbol{w}_{s_{j-1}}$ are the weights that belong to $\mathbf{F}_{s_j}$ but not to $\mathbf{F}_{s_{j-1}}$, $\boldsymbol{w}^{t+1}$ the global weights at communication round $t + 1$, $\boldsymbol{w}^{(i,t,E)}$ the weights on client $i$ at communication round $t$ after $E$ local iterations, $\mathcal{S}_t^j = \{i \in \mathcal{S}_t : p_{\max}^i \geq s_j\}$ a set of clients that have the capacity to update $\boldsymbol{w}_{s_j}$, and WA stands for weighted average, where weights are proportional to the amount of data on each client.

**Communication Savings.** In addition to the computational savings (§3.4), OD provides additional *communication* savings. First, for the server-to-client transfer, every device with $p_{\max}^i < 1$ observes a reduction of approximately $1/(p_{\max}^i)^2$ in the downstream transferred data due to the smaller model size (§ 3.4). Accordingly, the upstream client-to-server transfer is decreased by $1/(p_{\max}^i)^2$ as only the gradient updates of the unpruned units are transmitted.

**Algorithm 1: FjORD** (Proposed Framework)

---

**Input:** $\mathbf{F}, \boldsymbol{w}^0, D_{\mathcal{P}}, T, E$

1 **for** $t \leftarrow 0$ **to** $T - 1$ **do** // *Global rounds*
2     Server selects clients as a subset $\mathcal{S}_t \subset \mathcal{A}_t$
3     Server broadcasts weights of $p_{\max}^i$-submodel to each client $i \in \mathcal{S}_t$
4     **for** $k \leftarrow 0$ **to** $E - 1$ **do** // *Local iterations*
5         $\forall i \in \mathcal{S}_t$: Device $i$ samples $p_{(i,k)} \sim D_{\mathcal{P}} | D_{\mathcal{P}} \leq p_{\max}^i$ and updates the weights of local model
6     **end**
7     $\forall i \in \mathcal{S}_t$: device $i$ sends to the server the updated weights $\boldsymbol{w}^{(i,t,E)}$
8     Server updates $\boldsymbol{w}^{t+1}$ as in Eq. (1)
9 **end**

---

**Identifiability.** A standard procedure in FL is to perform element-wise averaging to aggregate model updates from clients. However, coordinate-wise averaging of updates may have detrimental effects on the accuracy of the global model, due to the permutation invariance of the hidden layers. Recent techniques tackle this problem by matching clients' neurons before averaging [68, 57, 62]. Unfortunately, doing so is computationally expensive and hurts scalability. FjORD mitigates this issue since it exhibits the natural importance of neurons/channels within each hidden layer by design; essentially OD acts in lieu of a neuron matching algorithm without the computational overhead.

**Subnetwork Knowledge Transfer.** In § 3.2, we introduced knowledge distillation for our OD formulation. We extend this approach to FjORD, where instead of the full network, we employ width $\max\{p \in \mathcal{P} : p \leq p_{\max}^i\}$ as a teacher network in each local iteration on device $i$. We provide the alternative of FjORD *with* knowledge distillation mainly as a solution for cases where the client bottleneck is memory- or network-related, rather than computational in nature [32]. However, in cases where client devices are computationally bound in terms of training latency, we propose FjORD *without* KD or decreasing $p_{\max}^i$ to account for the overhead of KD.

## 5 Evaluation of FjORD

In this section, we provide a thorough evaluation of FjORD and its components across different tasks, datasets, models and device cluster distributions to show its performance, elasticity and generality.

**Datasets and Models.** We evaluate FjORD on two vision and one text prediction task, shown in Table 1a. For CIFAR10 [33], we use the "CIFAR" version of ResNet18 [20]. We federate the dataset by randomly dividing it into equally-sized partitions, each allocated to a specific client, and thus remaining IID in nature. For FEMNIST, we use a CNN with two convolutional layers followed by a softmax layer. For Shakespeare, we employ a RNN with an embedding layer (without dropout) followed by two LSTM [22] layers and a softmax layer. We report the model's performance of the last epoch on the test set which is constructed by combining the test data for each client. We report top-1 accuracy vision tasks and negative perplexity for text prediction. Further details, such as hyperparameters, description of datasets and models are available in the Appendix.

Table 1: Datasets and models

|  | $p = 0.2$ | 0.4 | 0.6 | 0.8 | 1.0 |
|---|---|---|---|---|---|
| **CIFAR10 / ResNet18** | | | | | |
| MACs | 23M | 91M | 203M | 360M | 555M |
| Params | 456K | 2M | 4M | 7M | 11M |
| **FEMNIST / CNN** | | | | | |
| MACs | 47K | 120K | 218K | 342K | 491K |
| Params | 5K | 10K | 15K | 20K | 26K |
| **Shakespeare / RNN** | | | | | |
| MACs | 12K | 40K | 83K | 143K | 216K |
| Params | 12K | 40K | 82K | 142K | 214K |

| Dataset | Model | # Clients | # Samples | Task |
|---|---|---|---|---|
| CIFAR10 | ResNet18 | 100 | 50,000 | Image classification |
| FEMNIST | CNN | 3,400 | 671,585 | Image classification |
| Shakespeare | RNN | 715 | 38,001 | Next character prediction |

(a) Datasets description

(b) MACs and parameters per $p$-reduced network

### 5.1 Experimental Setup

**Infrastructure.** FjORD was implemented on top of the `Flower` (v0.14dev) [5] framework and PyTorch (v1.4.0) [51]. We run all our experiments on a private cloud cluster, consisting of Nvidia V100 GPUs. To scale to hundreds of clients on a single machine, we optimized `Flower` so that clients only allocate GPU resources when actively participating in a federated client round. We report average performance and the standard deviation across three runs for all experiments. To model client availability, we run up to 100 `Flower` clients in parallel and sample 10% at each global round, with the ability for clients to switch identity at the beginning of each round to overprovision for larger federated datasets. Furthermore, we model client heterogeneity by assigning each client to one of the device clusters. We provide the following setups:

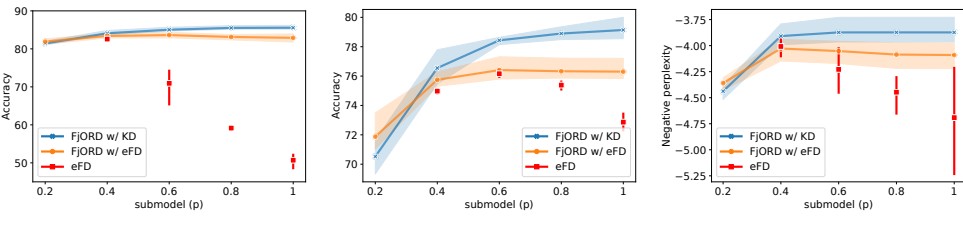

| (a) ResNet18 - CIFAR10 | (b) CNN - FEMNIST | (c) RNN - Shakespeare |

Figure 4: Ordered Dropout with KD vs eFD baselines. Performance vs dropout rate $p$ across different networks and datasets. $D_{\mathcal{P}} = \mathcal{U}_5$

**Uniform-{5,10}:** This refers to the distribution $D_{\mathcal{P}}$, *i.e.* $p \sim \mathcal{U}_k$, with $k = 5$ or $10$.

**Drop Scale** $\in \{0.5, 1.0\}$**:** This parameter affects a possible skew in the number of devices per cluster. It refers to the drop in clients per cluster of devices, as we go to higher $p$'s. Formally, for *uniform-n* and drop scale $ds$, the high-end cluster $n$ contains $1 - \sum_{i=0}^{n-1} ds/n$ of the devices and the rest of the clusters contain $ds/n$ each. Hence, for $ds=1.0$ of the *uniform-5* case, all devices can run the $p = 0.2$ subnetwork, 80% can run the $p = 0.4$ and so on, leading to a device distribution of $(0.2, ..., 0.2)$. This percentage drop is half for the case of $ds=0.5$, resulting in a larger high-end cluster, *e.g.* $(0.1, 0.1, ..., 0.6)$.

**Baselines.** To assess the performance against the state-of-the-art, we compare FjORD with the following baselines: i) Extended Federated Dropout (eFD), ii) FjORD with eFD (FjORD w/ eFD).

eFD builds on top of the technique of Federated Dropout (FD) [8], which adopts a Random Dropout (RD) at neuron/filter level for minimising the model's footprint. However, FD does not support adaptability to heterogeneous client capabilities out of the box, as it inherits a *single* dropout rate across devices. For this reason, we propose an extension to FD, allowing to adapt the dropout rate to the device capabilities, defined by the respective cluster membership. It is clear that eFD dominates FD in performance and provides a tougher baseline, as the latter needs to impose the same dropout rate to fit the model at hand on all devices, leading to larger dropout rates (*i.e.* uniform dropout of 80% for full model to support the low-end devices). We provide empirical evidence for this in the Appendix. For investigative purposes, we also applied eFD on top of FjORD, as a means to update a larger part of the model from lower-tier devices, *i.e.* allow them to evaluate submodels beyond their $p_{\max}^i$ during training.

## 5.2 Performance Evaluation

In order to evaluate the performance of FjORD, we compare it to the two baselines, eFD and OD+eFD. We consider the *uniform-5* setup with drop scale of 1.0 (*i.e.* uniform clusters). For each baseline, we train one independent model $\mathbf{F}_p$, end-to-end, for each $p$. For eFD, what this translates to is that the clusters of devices that cannot run model $\mathbf{F}_p$ compensate by randomly dropping out neurons/filters. We point out that $p = 0.2$ is omitted from the eFD results as it is essentially not employing any dropout whatsoever. For the case of FjORD + eFD, we control the RD by capping it to $d = 0.25$. This allows for larger submodels to be updated more often – as device belonging to cluster $c$ can now have $p_{\max}^c \rightarrow p_{\max}^{c+1}$ during training where $c+1$ is the next more powerful cluster – while at the same time it prevents the destructive effect of too high dropout values shown in the eFD baseline.

Fig. 4 presents the achieved accuracy for varying values of $p$ across the three target datasets. FjORD (denoted by FjORD w/ KD) outperforms eFD across all datasets with improvements between 1.53-34.87 percentage points (pp) (19.22 pp avg. across $p$ values) on CIFAR10, 1.57-6.27 pp (3.41 pp avg.) on FEMNIST and 0.01-0.82 points (p) (0.46 p avg.) on Shakespeare. Compared to FjORD +eFD, FjORD achieves performance gains of 0.71-2.66 pp (1.79 avg.), up to 2.56 pp (1.35 pp avg.) on FEMNIST and 0.12-0.22 p (0.18 p avg.) on Shakespeare.

Across all tasks, we observe that FjORD is able to improve its performance with increasing $p$ due to the nested structure of its OD method. We also conclude that eFD on top of FjORD does not seem to lead to better results. More importantly though, given the heterogeneous pool of devices, to obtain the highest performing model for eFD, multiple models have to be trained (*i.e.* one per device cluster). For instance, the highest performing models for eFD are $\mathbf{F}_{0.4}$, $\mathbf{F}_{0.6}$ and $\mathbf{F}_{0.4}$ for CIFAR10, FEMNIST and Shakespeare respectively, which can be obtained only *a posteriori*; after all model variants have been trained. Instead, despite the device heterogeneity, FjORD requires a single training process that leads to a global model that significantly outperforms the best model of eFD (by 2.98 and 2.73 pp for CIFAR10 and FEMNIST, respectively, and 0.13 p for Shakespeare), while allowing

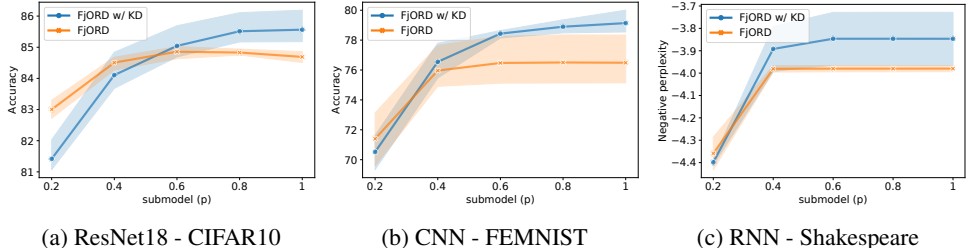

|(a) ResNet18 - CIFAR10|(b) CNN - FEMNIST|(c) RNN - Shakespeare|

Figure 5: Ablation analysis of FjORD with Knowledge Distillation. Ordered Dropout with $D_{\mathcal{P}} = \mathcal{U}_5$, KD - Knowledge distillation.

the direct, seamless extraction of submodels due to the nested structure of OD. Empirical evidence of the convergence of FjORD and the corresponding baselines is provided in the Appendix.

### 5.3 Ablation Study of KD in FjORD

To evaluate the contribution of our *knowledge distillation* method to the attainable performance of FjORD, we conduct an ablative analysis on all three datasets. We adopt the same setup of *uniform-5* and drop scale $= 1.0$ as in the previous section and compare FjORD with and without KD.

Fig. 5 shows the efficacy of FjORD's KD in FL settings. FjORD's KD consistently improves the performance across all three datasets when $p > 0.4$, with average gains of $0.18, 0.68$ and $0.87$ pp for submodels of size $0.6, 0.8$ and $1$ on CIFAR-10, $1.96, 2.39$ and $2.65$ pp for FEMNIST and $0.10$ p for Shakespeare. For the cases of $p \leq 0.4$, the impact of KD is fading. We believe this to be a side-effect of optimising for the average accuracy across submodels, which also yielded the $T = \alpha = 1$ strategy. We leave the exploration of alternative weighted KD strategies as future work. Overall, the use of KD significantly improves the performance of the global model, yielding gains of $0.71$ and $2.63$ pp for CIFAR10 and FEMNIST and $0.10$ p for Shakespeare.

### 5.4 FjORD's Deployment Flexibility

#### 5.4.1 Device Clusters Scalability

An important characteristic of FjORD is its ability to *scale* to a larger number of device clusters or, equivalently, perform well with higher granularity of $p$ values. To illustrate this, we test the performance of OD across two setups, *uniform-5* and *-10* (defined in § 5.1).

As shown in Fig. 6, FjORD sustains its performance even under the higher granularity of $p$ values. This means that for applications where the modelling of clients needs to be more fine-grained, FjORD can still be of great value, without any significant degradation in achieved accuracy per submodel. This further supports the use-case where device-load needs to be modelled explicitly in device clusters (*e.g.* modelling device capabilities and load with deciles).

#### 5.4.2 Adaptability to Device Distributions

In this section, we make a similar case about FjORD's *elasticity* with respect to the allocation of available devices to each cluster. We adopt the setup of *uniform-5* once again, but compare across drop scales $0.5$ and $1.0$ (defined in § 5.1). In both cases, clients that can support models of $p_{\max}^i \in \{0.2, \dots, 0.8\}$ are equisized, but the former halves the percentage of devices and allocates it to the last (high-end) cluster, now accounting for $60\%$ of the devices. The rationale behind this is that the majority of participating devices are able to run the whole original model.

The results depicted in Fig. 7 show that the larger submodels are expectedly more accurate, being updated more often. However, the same graphs also indicate that FjORD does not significantly degrade the accuracy of the smaller submodels in the presence of more high-tier devices (*i.e.* $ds = 0.5$). This is a direct consequence of sampling $p$ values during local rounds, instead of tying each tier with only the maximal submodel it can handle. We should also note that we did not alter the uniform sampling in this case on the premise that high-end devices are seen more often, precisely to illustrate FjORD's adaptability to latent user device distribution changes of which the server may not be aware.

## 6 Related Work

**Dropout Techniques.** Contrary to conventional Random Dropout [59], which stochastically drops a different, random set of a layer's units in every batch and is typically applied for regularisation purposes, OD employs a *structured* ordered dropping scheme that aims primarily at tunably reducing the computational and memory cost of training and inference. However, OD can still have an implicit regularisation effect since we encourage learning towards the top-ranked units (*e.g.* the left-most units in the example of Fig. 2), as these units will be dropped less often during training. Respectively,

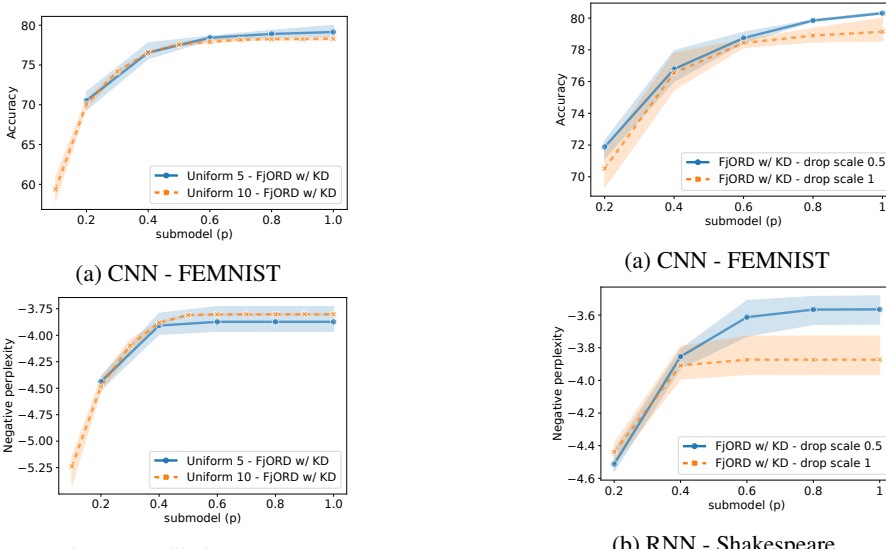

(a) CNN - FEMNIST

(a) CNN - FEMNIST

(b) RNN - Shakespeare

(b) RNN - Shakespeare

Figure 6: Demonstration of FjORD's scalability with respect to the number of device clusters.

Figure 7: Demonstration of the adaptability of FjORD across different device distributions.

at inference time, the load of a client can be dynamically adjusted by dropping the least important units, *i.e.* adjusting the width of the network.

To the best of our knowledge, the only similar technique to OD is Nested Dropout, where the authors proposed a similar construction, which is applied to the representation layer in autoencoders [54] in order to enforce identifiability of the learned representation or the last layer of the feature extractor [24] to learn an ordered set of features for transfer learning. In our case, we apply OD to every layer to elastically adapt the computation and memory requirements during training and inference.

**Traditional Pruning.** Conventional non-FL compression techniques can be applicable to reduce the network size and computation needs. The majority of pruning methods [18, 16, 37, 35, 49] aim to generate a *single* pruned model and require access to labelled data in order to perform a costly fine-tuning/calibration for *each* pruned variant. Instead, FjORD's Ordered Dropout enables the deterministic extraction of *multiple* pruned models with *varying* resource budgets directly after training. In this manner, we remove both the excessive overhead of fine-tuning and the need for labelled data availability, which is crucial for real-world, privacy-aware applications [60, 56]. Finally, other model compression methods [13, 64, 9] remain orthogonal to FjORD.

**System Heterogeneity.** So far, although substantial effort has been devoted to alleviating the *statistical heterogeneity* [38] among clients [58, 36, 26, 12, 40], the *system heterogeneity* has largely remained unaddressed. Considering the diversity of client devices, techniques on client selection [50] and control of the per-round number of participating clients and local iterations [45, 65] have been developed. Nevertheless, as these schemes are restricted to allocate a uniform amount of work to each selected client, they either limit the model complexity to fit the lowest-end devices or exclude slow clients altogether. From an aggregation viewpoint, [39] allows for partial results to be integrated to the global model, thus enabling the allocation of different amounts of work across heterogeneous clients. Despite the fact that each client is allowed to perform a different number of local iterations based on its resources, large models still cannot be accommodated on the more constrained devices.

**Communication Optimisation.** The majority of existing work has focused on tackling the communication overhead in FL. [32] proposed using structured and sketched updates to reduce the transmitted data. ATOMO [61] introduced a generalised gradient decomposition and sparsification technique, aiming to reduce the gradient sizes communicated upstream. [17] adaptively select the gradients' sparsification degree based on the available bandwidth and computational power. Building upon gradient quantisation methods [44, 23, 53, 25], [2] proposed using quantisation in the model sharing and aggregation steps. However, their scheme requires the *same* clients to participate across all rounds, and is, thus, unsuitable for realistic settings where clients' availability cannot be guaranteed. Despite the bandwidth savings, these communication-optimising approaches do not offer computational gains nor do they address device heterogeneity. Nonetheless, they remain orthogonal to our work and can be complementarily combined to further alleviate the communication cost.

**Computation-Communication Co-optimisation.** A few works aim to co-optimise both the computational and bandwidth costs. PruneFL [29] proposes an unstructured pruning method. Despite the similarity to our work in terms of pruning, this method assumes a *common* pruned model across *all* clients at a given round, thus not allowing more powerful devices to update more weights. Hence, the pruned model needs to meet the constraints of the least capable devices, which severely limits the model capacity. Moreover, the adopted unstructured sparsity is difficult to translate to processing speed gains [67]. Federated Dropout [8] randomly sparsifies the global model, before sharing it to the clients. Similarly to PruneFL, Federated Dropout does not consider the system diversity and distributes the *same* model size to all clients. Thus, it is restricted by the low-end devices or excludes them altogether from the FL process. Additionally, Federated Dropout does not translate to computational benefits at inference time, since the whole model is deployed after federated training.

Contrary to the presented works, our framework embraces the client heterogeneity, instead of treating it as a limitation, and thus pushes the boundaries of FL deployment in terms of fairness, scalability and performance by tailoring the model size to the device at hand, both at training and inference time, in a "train-once-deploy-everywhere" manner.

## 7    Conclusions & Future Work

In this work, we have introduced FjORD, a federated learning method for heterogeneous device training. To this direction, FjORD builds on top of our Ordered Dropout technique as a means to extract submodels of smaller footprints from a main model in a way where training the part also participates in training the whole. We show that our Ordered Dropout is equivalent to SVD for linear mappings and demonstrate that FjORD's performance in the local and federated setting exceeds that of competing techniques, while maintaining flexibility across different environment setups.

In the future, we plan to investigate how FjORD can be deployed and extended to future-gen devices and models in a life-long manner, the interplay between system and data heterogeneity for OD-based personalisation as well as alternative dynamic inference techniques for tackling system heterogeneity.

## Broader Impact

Our work has a dual broader societal impact: i) on privacy and fairness in participation and ii) on the environment. On the one hand, centralised DNN training [19] has been the norm for a long time, mainly facilitated by the advances in server-grade accelerator design and cheap storage. However, this paradigm comes with a set of disadvantages, both in terms of data privacy and energy consumption. With mobile and embedded devices becoming more capable and FL becoming a viable alternative [3, 7], one can leverage the free compute cycles of client devices to train models on-device, without data ever leaving the device premises. These devices, being typically battery-powered, operate under a more constrained power envelope compared to data-center accelerators [1]. Moreover, these devices are already deployed in the wild, but typically not used for training purposes. What FjORD contributes is the ability for even less capable devices to participate in the training process, thus increasing the representation of low-tier devices (and by extension the correlated demographic groups), as well as adding to the overall compute capabilities of the distributed system as a whole, potentially offsetting part of the carbon footprint of centralised training data centers [52].

However, moving the computation cost from the service provider to the user of the device is a non-negligible step and the user should be made aware what their device is used for, especially if they are contributing to the knowledge of a model they do not own. Moreover, while many large data centers [11, 15] are increasingly dependent on renewable resources for meeting their power demands, this might not be the case for household electricity, which may impede the sustainability of training on device, at least in the short run.

## Funding Disclosure

This work was entirely performed at and funded by Samsung AI.

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
