# Supplementary Material

## A   Proof of Theorem 1

*Proof.* Let $A = \hat{U}\Sigma\hat{V}^\top = \sum_{i=1}^k \sigma_i \hat{u}_i \hat{v}_i^\top$ denote SVD decomposition of $A$. This decomposition is unique as $A$ is full rank with distinct singular values. We also denote $A_b = \sum_{i=1}^b \sigma_i \hat{u}_i \hat{v}_i^\top$

Assuming a linkage of input data $x$ and response $y$ through a linear mapping $Ax = y$, we obtain the following

$$\min_{U,V} \mathbb{E}_{x \sim \mathcal{X}} \mathbb{E}_{p \sim D_\mathcal{P}} \|\mathbf{F}_p(x) - y\|^2 = \min_{U,V} \mathbb{E}_{x \sim \mathcal{X}} \mathbb{E}_{p \sim D_\mathcal{P}} \|\mathbf{F}_p(x) - Ax\|^2.$$

Let us denote $u_i$ to be $i$-th row of matrix U and $v_i^\top$ to be $i$-th row of V. Due to $\mathcal{X}$ being uniform on unit ball and structure of the neural network, we can further simplify the objective to

$$\min_{U,V} \mathbb{E}_{p \sim D_\mathcal{P}} \left\| \sum_{i=1}^{\lceil p \cdot k \rceil} u_i v_i^\top - A \right\|_F^2,$$

where $F$ denotes Frobenius norm. Since for each $b$ there exists nonzero probability $\mathbf{P}_b$ such that $b = \lceil p \cdot k \rceil$, we can explicitly compute expectation, which leads to

$$\min_{U,V} \sum_{b=1}^k \mathbf{P}_b \left\| \sum_{i=1}^b u_i v_i^\top - A \right\|_F^2.$$

Realising that $\sum_{i=1}^b u_i v_i^\top$ has rank at most $b$, we can use Eckart–Young theorem which implies that

$$\min_{U,V} \sum_{b=1}^k \mathbf{P}_b \left\| \sum_{i=1}^b u_i v_i^\top - A \right\|_F^2 \geq \sum_{b=1}^k \mathbf{P}_b \|A_b - A\|_F^2.$$

Equality is obtained if and only if $A_b = \sum_{i=1}^b u_i v_i^\top$ for all $i \in \{1, 2, \ldots, k\}$. This can be achieved, *e.g.* $v_i = \hat{v}_i$ and $u_i = \sigma_i \hat{u}_i$ for all $i \in \{1, 2, \ldots, k\}$.  □

## B   OD: Optimization Perspective

In this section, we discuss the impact of introducing the Ordered Dropout formulation into the original problem. We follow notation used in the main paper, where $\mathbf{F}$ and $w$ are the original network and weights, respectively, and $\mathbf{F}_p$ denotes a pruned $p$-subnetwork with its weights $w_p$. We argue that the problem does not become harder from the optimization point of view as quantities such as smoothness or strong convexity do not worsen for the OD formulation as stated in the following lemma.

**Lemma 1.** *Let $\mathbf{F}$ be $\mu$-strongly convex and $L$-smooth. Then $\mathbf{F}_{D_\mathcal{P}} = \mathbb{E}_{p \sim D_\mathcal{P}}[\mathbf{F}_p]$ is $\mu'$-strongly convex and $L'$-smooth with $\mu' \geq \mu$ and $L' \leq L$.*

*Proof.* The claim trivially follows from the definitions of smoothness and strong convexity by realising that $\mathbf{F}_p(w_p)$ is equal to $\mathbf{F}(w)$ where $w_p$ is obtained from $w$ using our pruning technique, thus $\mathbf{F}_p$ is $\mu_p$-strongly convex and $L_p$-smooth with $\mu_p \geq \mu$ and $L_p \leq L$ for all $p \in [0, 1]$. Subsequently, the same has to hold for the expectation.  □

## C   Experimental Details

### C.1   Datasets and Models

Below, we provide detailed description of the datasets and models used in this paper. We use vision datasets EMNIST [11] and its federated equivalent FEMNIST and CIFAR10 [36], as well as the language modelling dataset Shakespeare [49]. In the centralised training scenarios, we use the union

of dataset partitions for training and validation, while in the federeated deployment, we adopt either a random partitioning in IID datasets or the pre-partitioned scheme available in TensorFlow Federated (TFF) [4]. Detailed description of the datasets is provided below.

**CIFAR10.** The CIFAR10 dataset is a computer vision dataset consisting of $32 \times 32 \times 3$ images with $10$ possible labels. For federated version of CIFAR10, we randomly partition dataset among $100$ clients, each client having $500$ data-points. We train a ResNet18 [23] on this dataset, where for Ordered Dropout, we train independent batch normalization layers for every $p \in D_{\mathcal{P}}$ as Ordered Dropout affects distribution of layers' outputs. We perform standard data augmentation and preprocessing, *i.e.* a random crop to shape $(24, 24, 3)$ followed by a random horizontal flip and then we normalize the pixel values according to their mean and standard deviation.

**(F)EMNIST.** EMNIST consists of $28 \times 28$ gray-scale images of both numbers and upper and lower-case English characters, with $62$ possible labels in total. The digits are partitioned according to their author, resulting in a naturally heterogeneous federated dataset. EMNIST is collection of all the data-points. We do not use any preprocessing on the images. We train a Convolutional Neural Network (CNN), which contains two convolutional layers, each with $5 \times 5$ kernels with $10$ and $20$ filters, respectively. Each convolutional layer is followed by a $2 \times 2$ max pooling layer. Finally, the model has a dense output layer followed by a softmax activation. FEMNIST refers to the federated variant of the dataset, which has been partitioned based on the writer of the digit/character [9].

**Shakespeare.** Shakespeare dataset is also derived from the benchmark designed by [9]. The dataset corpus is the collected works of William Shakespeare, and the clients correspond to roles in Shakespeare's plays with at least two lines of dialogue. Non-federated dataset is constructed as a collection of all the clients' data-points in the same way as for FEMNIST. For the preprocessing step, we apply the same technique as TFF dataloader, where we split each client's lines into sequences of $80$ characters, padding if necessary. We use a vocabulary size of $90$ entities – $86$ characters contained in Shakespeare's work, beginning and end of line tokens, padding tokens, and out-of-vocabulary tokens. We perform next-character prediction on the clients' dialogue using an Recurrent Neural Network (RNN). The RNN takes as input a sequence of $80$ characters, embeds it into a learned $8$-dimensional space, and passes the embedding through two LSTM [25] layers, each with $128$ units. Finally, we use a softmax output layer with $90$ units. For this dataset, we don't apply Ordered Dropout to the embedding layer, but only to the subsequent LSTM layers, due to its insignificant impact on the size of the model.

## C.2    Implementation Details

FjORD was built on top of `PyTorch` [54] and `Flower` [6], an open-source federated learning framework which we extended to support Ordered, Federated, and Adaptive Dropout and Knowledge Distillation. Our OD aggregation was implemented in the form of a `Flower` strategy that considers each client maximum width $p_{\max}^i$. Server and clients run in a multiprocess setup, communicating over gRPC[7] channels and can be distributed across multiple devices. To scale to hundreds of clients per cloud node, we optimised `Flower` so that clients only allocate GPU resources when actively participating in a federated client round. This is accomplished by separating the forward/backward propagation of clients into a separate spawned process which frees its resources when finished. Timeouts are also introduced in order to limit the effect of stragglers or failed client processes to the entire training round.

## C.3    Hyperparameters.

In this section we lay out the hyperparameters used for each $\langle$model, dataset, deployment$\rangle$ tuple.

### C.3.1    Non-federated Experiments

For centralised training experiments, we employ `SGD` with momentum $0.9$ as an optimiser. We also note that the training epochs of this setting are significantly fewer that the equivalent federated training rounds, as each iteration is a full pass over the dataset, compared to an iteration over the sampled clients.

---

[7]`https://www.grpc.io/`

**ResNet18.** We use batch size $128$, step size of $0.1$ and train for $300$ epochs. We decrease the step size by a factor of $10$ at epochs $150$ and $225$.

**CNN.** We use batch size $128$ and train for $20$ epochs. We keep the step size constant at $0.1$.

**RNN.** We use batch size $32$ and train for $50$ epochs. We keep the step size constant at $0.1$.

### C.3.2 Federated Experiments

For each $\langle$model, dataset$\rangle$ federated deployment, we start the communication round by randomly sampling $10$ clients to model client availability and for each available client we run one local epoch. We decrease the client step size by $10$ at $50\%$ and $75\%$ of total rounds. We run $500$ global rounds of training across experiments and use SGD without momentum.

**ResNet18.** We use local batch size $32$ and step size of $0.1$.

**CNN.** We use local batch size $16$ and step size of $0.1$.

**RNN.** We use local batch size $4$ and step size of $1.0$.

## D  Additional Experiments

**Ordered Dropout exactly recovers SVD: Empirical evidence**

In this experiment, we want to empirically back up our theoretical claims about Ordered Dropout recovering SVD. To this end, we generate a normal random $5 \times 5$ matrix, compute its SVD ($USV^T$) and set $A = UDV^T$ where D is a diagonal matrix with $5, 4, \ldots, 1$ on the diagonal to ensure full rank with distinct singular values. We initialize $U, V$ (1st & 2nd layer, see Theorem 1's proof) as normal random matrices. We then run SGD with lr=0.1 for 10k iterations. In each step, we sample 32 points from the 5D unit ball and apply OD sampled from the uniform distribution, taking an SGD step. In Fig. 8, we display the Frobenius norm between the best $k$-rank approximation of $A$ and the corresponding OD subnetwork ($p = [0.2, \ldots, 1]$, for $k = 1, \ldots, 5$). This converges and recovers SVD.

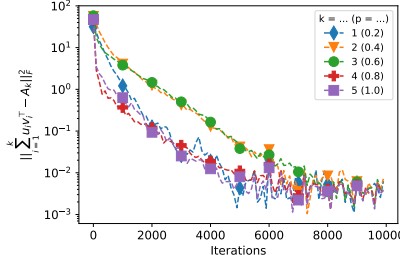

Figure 8: Frobenius norm of best $k$-rank approximation vs. corresponding OD sub-network.

**Local training: Ordered Dropout vs. Random Dropout vs. Single Models**

In this section, we present an extended version of Fig. 3, depicted in Fig. 9. We expand the comparison of OD against Random Dropout, where we pick the original model (SM1) and utilise different dropout rates for obtaining submodels. It can be easily witnessed that performance drops catastrophically in the latter case.

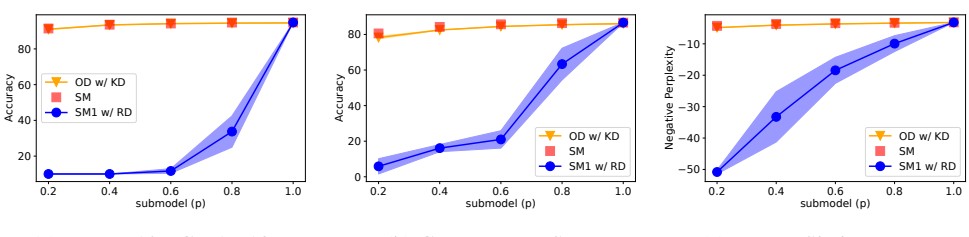

(a) ResNet18 - CIFAR10      (b) CNN - EMNIST      (c) RNN - Shakespeare

Figure 9: Full non-federated datasets. OD-Ordered Dropout with $D_{\mathcal{P}} = \mathcal{U}_5$, SM-single independent models, KD-knowledge distillation, SM1 w/ RD-end-to-end trained full model with submodels obtained using Random Dropout instead Ordered Dropout.

**Convergence of FjORD**

In this section, we depict the convergence behaviour of FjORD compared to the eFD baseline across 500 global training rounds. We follow the same setup as in § 5.2. From Fig. 10, it can be easily witnessed that FjORD (with or without distillation) leads to smoother convergence and yields lower losses across the three (model,dataset) combinations.

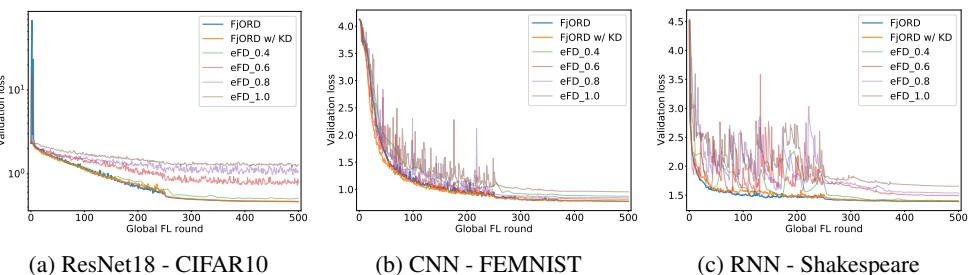

| (a) ResNet18 - CIFAR10 | (b) CNN - FEMNIST | (c) RNN - Shakespeare |

Figure 10: Convergence of FjORD vs. eFD over 500 FL training rounds.

**Federated Dropout vs. eFD**

In this section we provide evidence of eFD's accuracy dominance over FD. We inherit the setup of the experiment in § 5.2 to be able to compare results and extrapolate across similar conditions. From Fig. 11, it is clear that eFD's performance dominates the baseline FD by 27.13-33 percentage points (pp) (30.94 pp avg.) on CIFAR10, 4.59-9.04 pp (7.13 pp avg.) on FEMNIST and 1.51-6.96 points (p) (3.96 p avg.) on Shakespeare. The superior performance of eFD, as a technique, can be attributed to the fact that it allows for an adaptive dropout rate based on the device capabilities. As such, instead of imposing a uniformly high dropout rate to accommodate the low-end of the device spectrum, more capable devices are able to update larger portion of the network, thus utilising its capacity more intelligently.

However, it should be also noted that despite FD's accuracy drop, on average it is expected to have a lower computation/upstream network bandwidth/energy impact on devices of the higher end of the spectrum, as they run the largest dropout rate possible to accommodate the computational need of their lower-end counterparts. This behaviour, however, can also be interpreted as wasted computation potential on the higher end – especially under unconstrained environments (*i.e.* device charging overnight) – at the expense of global model accuracy.

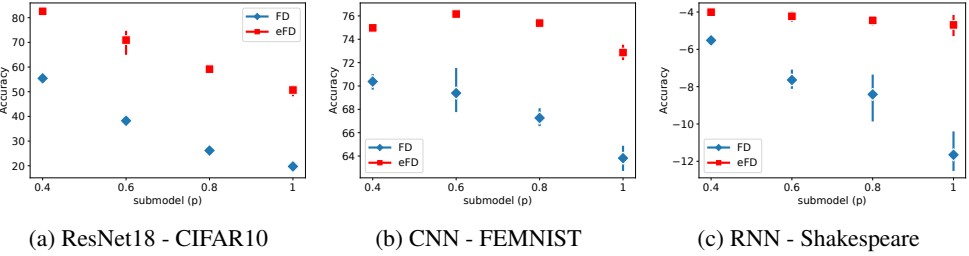

| (a) ResNet18 - CIFAR10 | (b) CNN - FEMNIST | (c) RNN - Shakespeare |

Figure 11: Federated Dropout (FD) vs Extended Federated Dropout (eFD). Performance vs dropout rate $p$ across different networks and datasets.

# E Limitations

In this work we have presented a method for training DNNs in centralised (through OD) and federated (through FjORD) settings. We have evaluated our method in terms of accuracy performance against baselines and enhancements of those that represent the state-of-the-art at the time of writing across three different datasets and tasks in IID and non-IID settings. Our evaluation has adopted uniform sampling of $p$ values and considered to be constant across layers of the network at hand mainly

for simplicity and tractability of the problem. However, this does not degrade the generality of our approach.

While we do target heterogeneous devices found in the wild, such as mobile phones, we have not measured the performance of our technique on such devices, mainly due to the lack of maturity in tools for on-device training. However, we have demonstrated the performance gains in terms of FLOPs and parameters in Table 1b, which are directly correlated with on-device performance, memory footprint and communication size. We defer on-device benchmarking and in-the-wild deployment at scale for future work.

Last, we have assumed the clusters of devices to be given and the different device load to be assumed in the modelling of these clusters. While we do provide results across different number and distribution of clusters in § 5.4, we treat the device-to-cluster association outside the scope of this work.