# OpenReview forum: "FjORD: Fair and Accurate Federated Learning under heterogeneous targets with Ordered Dropout"
_NeurIPS.cc/2021/Conference — NeurIPS 2021 Spotlight_

### Official Review · Reviewer_7FTD · 2021-07-05

**Rating:** 8
**Confidence:** 5

**Summary:**

The paper proposed a scheme for training deep models in the setup of federated learning, where updates to different nested submodels can be trained at different clients with different hardware capabilities.

(+) I appreciate solid grounding in practical motivation with potential significant practical value, and I would love to see further works in this direction.

(-) I feel execution on these ideas is not good enough, especially in terms of matching empirical baselines.

**Limitations And Societal Impact:**

Yes

**Main Review:**

### Presentation

Overall, the paper reads well, clearly describes relation to prior work, and its contributions. I only have minor comments and suggestions for improvement.

One detail in presentation I found confusing is in Sec 1-3 is whether the target impact of this work is on training or deployment for inference. In some way both, but I think these two points are somewhat confusingly entangled and may be worth explicitly separating as two distinct values delivered.

In Fig 2, I find the visualization misleading. The two diagrams for p=0.2 are networks functionally equivalent with respect to permutation of activations. I believe the crucial feature of proposed approach is the relation in line 113, and should be promoted in the narrative. I see it as the main originality / novelty, and while the idea may seem simple, I think it is very important.

Lines 81-83 I don't get the comment about computational benefits. No matter how random neurons are sampled, in the setup you have, you can still represent the model forwards / backward pass using dense matrix multiplies of smaller size.

Sec 3.1, the focus on layers when explaining the idea seems confusing to me, as it primarily impacts how adjacent layers interact. Moreover, the presentation seems to apply only to something that would fit into tf.keras.Sequential, but really is more general.

Sec 3.3 feels quite disruptive in the overall flow. I would either move it completely to appendix, leaving just a brief shoutout, or include more details about what this actually means. U, V are for instance not even defined, and I can't really understand what you try to say without looking at the appendix.

In Alg 1, server already makes a decision about what submodel to send to a client ( p_{max}^i ). If that is already the case, why not also include the sampling in the server's decision and save some communication? [I now see this would not quite work for KD variant]. Perhaps look up how could the story be improve with PIR here [your 31, Sec 4.2.1].

Eq (2) - I think this is a problem only if you are aggregating the updated models. If you instead aggregate the model updates (difference between initial and updated model), this would not be a problem (?) Similarly, paragraph in line 201+ seems not relevant afterwards (?)



Fig 4/5 have overlap in the same data (FjORD w/ KD) but different color. That is confusing.

### Experimental evaluation

Major problem I see is lack of focus on recovering solid baseline model quality.

In Fig 3. I would like to see additional baseline, which should make a strong case for the OD idea and highlight why line 113 is important. That is, take the SM p=1.0 model, and create random submodels from it for smaller p values. I would expect the model quality to be below the OD line. **Could you please include this in response?**

For CIFAR 10 dataset partitioning, I would like to see something else than uniform split. See perhaps "Reddi et al, Adaptive Federated Optimization" for ideas. But even then, I would be looking more at the other two datasets.

Somewhere, I would in more detail explain how to apply the idea to RNNs, I think that is not at all intuitive.

The major problem I see is in Fig 4/5. The accuracies are significantly below baselines (omitted from these figures). Baselines presented in Fig 2 in this paper (!) and also other works, for instance in mentioned "Reddi et al., Adaptive Federated Optimization". I see this as a significant problem for this work which is focused primarily on the practical side of things.

If you are able to almost match the baselines for p=1.0, and have this additional message that submodels are readily available, that would be a very strong message to me. As is, it appears as collection of regimes that do not quite work, yet.

### Ideas for improvement

I would mainly focus on narrowing the gap from existing baselines.

I like the messaging around device heterogeneity. What is completely missing, is that the device hardware is likely going to correlate with the data available to it. I think if you took a richer dataset, maybe stackoverflow which contains lot more information based on which to partition the data, you could create a setup when comparing full model trained on a single cluster, and FjORD trained model on two (or all) clusters. There is a chance that creating really strong contribution would be easier with such larger / richer dataset, than it is with the simple datasets used in the current submission.


**Time Spent Reviewing:**

3.5

---

> ### Author Response · Authors · 2021-08-10
> **Response to the review**
>
> We would like to thank the reviewer for their extensive feedback and insightful ideas for improvement. We hope that our response will clarify some things and prove to the reviewer that FjORD is not a "collection of regimes that do not quite work, yet".
>
> **wrt Clarity of Presentation.** We thank the reviewer for the suggestions on how to make the manuscript even clearer. We plan to address all the following points:
> 1) In Section 3.1, we will clarify that the value of $p$ affects the number of output channels and thus the number of input channels of the next layer;
> 2) We will address the color mismatch between Fig. 4 and 5.
> 3) We may move Section 3.3 in the Appendix.
>
> **wrt Fig. 2.** It is indeed correct that for $p=0.2$ the networks are functionally equivalent. The main difference is that for ordered dropout (OD) once $p=0.2$ is sampled, the pruning procedure is deterministic and yields a nested structure as seen in Fig. 2 (right). We highlight in the caption that in OD there is a natural ordering of the importance of these neurons, but we agree that the nested nature of the submodels is a major point to bring forward. We will attempt to address this in the visualisation with different colours and opacity levels.
>
> **wrt Section 3.3.** Thank you for the suggestion, we will incorporate this in the revised manuscript. The main message from this section is “Our OD formulation exhibits not only intuitively, but also theoretically ordered importance representation.” (lines 128-129) $U$ and $V$ stand for the first and second layer parameters, respectively. We apologise for the missing definition.
>
> **wrt Training vs Inference.** We *primarily* propose OD for (federated) training.
> However, an *additional advantage* is that these can be also deployed for tuning the model footprint at inference time. For training, we illustrate how the OD models behave and their convergence behaviour. We also show adaptability to different distributions of devices and allocations of data. For inference, we show the accuracies for each of the submodels (usable for inference) and the difference in parameters/FLOPs (inference) to quantify computational gains. Moreover, we also contrast OD vs RD for inference, with the latter being almost unusable in the case of submodel inference (see extra baseline). As such, while we do pose OD as a training mechanism for ordered representation of knowledge in a DNN, we also claim and showcase that this structure has benefits at inference time, too. We appreciate the comment for lack of clarity in the presentation and we plan to add a separate subsection discussing the inference deployment benefits of OD and as well as  “train-once-deploy-everywhere" nature (i.e. extract submodel without fine-tuning).
>
> **wrt Computational Benefits.**
> **TLDR**; Random Dropout at neuron level involves additional manual housekeeping to be done efficiently. OD with its nested representation and adjacent neuron pruning does not and thus is effectively more performant out of the box. (see further commentary on *"computational benefits of OD vs (e)FD"* for additional details)
>
> **wrt Sampling at the Server.** $p$ value sampling at the server-side (before downstream communication) would, in general, be possible, but would prevent us from
>  i) sampling p values at each local step (line 5, Alg.1) and
>  ii) using KD.
> If sampling happened on the server-side, then we intuitively expect it to lead to worse performance as the client update (which consists of *the full local epoch*) would be biased towards single sampled $p_i$ ignoring the rest of the distribution. Moreover, the client would not be able to use any of the submodels up to p^i_{\max} for inference until the next communication. Last, if the $p$ value also reflects on instantaneous device load, this would need to be communicated to the server, whereas in our case the device can simply bias it's sampling method towards lower p values under high load.
> We would also like to thank the reviewer for their private information retrieval suggestion. Indeed its inclusion would allow FjORD to run in a way that no client needs to reveal its capabilities/constraints. We leave such exploration as future work.
>
> **wrt aggregating Weights vs Updates**. Equation (2) is a technicality to avoid averaging updates/weights that do not exist because each client only communicates the model/update up to its maximal width $p^i_{\max}$. Therefore, equation (2) is needed regardless of whether clients communicate updates or models (we can do either one).  With regards to the identifiability paragraph, the same still applies and the presented reasoning applies regardless of whether clients communicate updates or models.
>
> **wrt Extra Baseline.** Thank you for this useful suggestion, we include such a baseline below.
>
> (
> [CIFAR10](https://drive.google.com/file/d/1CXTSoKSy7Vda7cXBPnkfeUZOKkFmGAuR/view?usp=sharing),
> [EMNIST](https://drive.google.com/file/d/1Ch9nQE9w4gZRVJLF5pDx9NfyRlKEl99T/view?usp=sharing),
> [Shakespeare](https://drive.google.com/file/d/1SJSSOH9yEMIayul--G07vWK2jR_gM5qz/view?usp=sharing)
> )
>
> As it can be noted, the structure of OD brings a significant improvement over this naive baseline, which further supports the importance of width-based dropout proposed in our work. We will update Figure 3 accordingly.
>
> **wrt Narrowing the Gap from Existing Baselines.** We would like to kindly disagree with the statements provided by the reviewer. Let us present our case.
> Firstly, compared to single node training (i.e. Figure 3 of our paper), a federated alternative is expected to have degraded performance due to the latter's challenging nature due to non-IID data and optimisation constraints. This is a fact not only affecting our work, but others including the mentioned work from Reddi et al.
> Indicatively, the best performance reported in the paper for CIFAR 10/100 in the FL setup for ResNet18 is $78.0$% and $52.5$%, respectively, while for one node setup it is easy to obtain accuracies of $\sim 94$% and $\sim 75$%, respectively. Therefore, one can think about a single node setting as an upper bound rather than a baseline that one expects to be surpassed/matched.
> Next, comparing FjORD with the results from "Adaptive Federated Optimization", one should look into experiments of similar setups (i.e. model architecture, dataset and communication rounds). The only setup in which we use the same model as Reddi et al. is ResNet18 for CIFAR10. While we only run $500$ communication rounds (Reddi et al. $4000$), our obtained accuracy of the full model is more than $85$% compared to the best $78$% for Reddi et al. We realise that this could be mainly because we use i.i.d data split while Reddi et al. use unbalanced split, also mentioned by the reviewer.
> To provide some additional meaningful comparisons, we conducted extra experiments, where we use the same setup as Reddi et al. for CIFAR100 (same dataset split as Reddi et al.) and Shakespeare dataset with the difference that clients are assigned into 5 clusters in the uniform 5 style (defined in Section 5.1). We use FedAvg with momentum as an optimizer and run $2000$ ($4000$ Reddi et al.) and $1000$ ($1200$ Reddi et al.) for CIFAR100 and Shakespeare, respectively. We visualize our results below. Surprisingly, our obtained accuracy for the full model is $52.62$% and $61.13$% while  Reddi et al. report  $52.5$% and $57.5$%  (Table 1, Adaptive Federated Optimization), respectively. This is despite the fact that we run fewer communication rounds, we don’t perform any tuning while Reddi et al. report the performance of the best model after an extensive search over the optimizers and step sizes, which one could do also for FjORD and thus one could expect even better performance for FjORD and we operate in a more challenging heterogeneous setup with clients of limited capabilities. We conjecture that this might be due to implicit regularization provided by our OD formulation.
>
> (
> [CIFAR100](https://drive.google.com/file/d/1X4AxaEMimlpiiGetR-kEIsmVR-X_57t1/view?usp=sharing),
> [Shakespeare](https://drive.google.com/file/d/1Sk407e9sgpOZ8cb8X0Yz6UX8yBrUZSsq/view?usp=sharing)
> )
>
> To summarise, we believe that FjORD's performance is on par or greater than federated baselines (with heterogeneous and homogeneous clients) and only below centralised training settings, which arguably hurt user privacy. As such, we conjecture that FjORD is indeed a viable solution to the problem of system heterogeneity.
>
> **wrt Ordered Dropout for RNN.** Standard RNN or LSTM cells can be seen as a set of linear feed-forward layers with activation and composition functions. We apply OD to these linear layers and sampled $p$ is fixed for a single recurrent forward-backward pass. We hope that this clarifies the reviewer’s concern. We will incorporate such a discussion in the manuscript to improve clarity.
>
> **wrt the Improvements.**
> Thank you for these valuable suggestions. We would like to add a subsection at the end of the paper about potential future directions of research, including the device-data heterogeneity correlation. We would be grateful if the reviewer had some specific work in mind exploring this direction of work.

---

> > ### Author Response · Authors · 2021-08-10
> > **Computational benefits of OD vs (e)FD**
> >
> > In more detail, let us clarify the sources of inefficiency of Random Dropout and explain the computational gains of our OD method.  Initially, we would like to differentiate between two dropout categories: i) weights dropout - where individual weights are dropped (either randomly or in a structured manner) and ii) neurons dropout - where individual neurons are dropped.
> >
> > In case i), dropping a weight leads to individual elements of the layers’ matrices becoming zero. These zero values are randomly distributed across the matrix, requiring a sparse matrix multiply operation. If the pruning/dropout is random, it is effectively non-structured. As such, instead of dense matrix multiplications, we have sparse matrix multiplications. Even if the sparse matrix multiplication is implemented as a blocked matrix multiply, i.e. decomposed into the matrix multiply of submatrices, the submatrices would again be sparse.
> >
> > However, in the paper, both Random and Ordered Dropout fall under the category ii). In this case, dropping a neuron corresponds to removing a whole row or column from the corresponding matrices. Specifically, in fully-connected (FC) and recurrent layers, when an output neuron is dropped, a whole row of the layer’s weights matrix is removed. Similarly, in the matrix multiplication formulation of convolutional (CONV) layers, a whole row of the input activations matrix is removed. As a result, the resulting matrices can be represented in a dense form. However, despite the dense form,  there are still hidden costs that introduce inefficiencies upon execution.
> >
> > The inefficiency of this approach stems from the fact that, when done randomly (Random Dropout), the underlying implementation needs both additional data structures and operations to correctly perform the matrix multiply and generate a correctly-shaped output tensor. This includes overhead in 1) storing the sparse representation (i.e. which rows/columns are not pruned), 2) extracting the right (non-pruned) rows and columns indices (i.e. decoding the sparse representation), 3) turning the sparse matrix to a dense new one, and 4) reconstructing the resulting matrix to match the original non-pruned shape after performing the matrix multiply operation. All these lead to non-optimised execution out of the box.
> >
> > Moreover, for case ii), It has been extensively demonstrated that CPUs and GPUs (i.e. the currently available commodity hardware) and existing libraries (i.e. the existing software support) have noticeably reduced performance when executing sparse matrix multiplications, compared to when executing dense matrix multiplications [1-4]. Please note that existing implementations may execute the sparse matrix multiplication as dense matrix multiplication, but this is wrapped within sparse matrix kernels that perform the four aforementioned steps before and after the actual multiplication, leading to reduced processing speed.
> >
> > Our Ordered Dropout method completely alleviates the sources of inefficiency of Random Dropout, through its nested pruning scheme that requires neither additional data structures for bookkeeping nor complex and costly data layout transformations. As such, it can capitalise directly over the existing and highly optimised dense matrix multiplication libraries.
> >
> >
> > [1] Wang, Ziheng. "SparseRT: Accelerating Unstructured Sparsity on GPUs for Deep Learning Inference." Proceedings of the ACM International Conference on Parallel Architectures and Compilation Techniques (PACT). 2020.
> >
> > [2] Guan, Yijin, et al. "Crane: Mitigating Accelerator Under-utilization Caused by Sparsity Irregularities in CNNs." IEEE Transactions on Computers (TC). 2020.
> >
> > [3] Baidu Research. “DeepBench.” 2020.  https://github.com/baidu-research/DeepBench
> >
> > [4] Zhang, Shijin, et al. "Cambricon-X: An Accelerator for Sparse Neural Networks." 2016 49th Annual IEEE/ACM International Symposium on Microarchitecture (MICRO). IEEE, 2016.

---

> > > ### Comment · Reviewer_7FTD · 2021-08-26
> > > **Response**
> > >
> > > Thank you for detailed reply. I may have initially rated the work harshly, and you make me confident I was too harsh, especially on the experimental side. I increase the overall rating and will argue for acceptance.
> > >
> > > Much appreciated the extra plots with Fig 3. I did not expect the gap to be that big! Good for your paper! Maybe best to just drop it in appendix, as the difference between SM and OD is not visible with expanded y-axis. Thank you also for the extra experiments and discussion.
> > >
> > > wrt the computational benefits section, I don't agree - but this is a detail that does not diminish the value of the paper.
> > > For random dropout, all of the 4 points of inefficiency you list, can be at the server, where I would not worry about that kind of computatoinal cost. Maybe if you look at it from the point of view of the client - it can just receive ML program and corresponding model weights, do its job in local training and sending back updates - without knowing anything about what is the model actually being trained at the server.

---

> > > > ### Author Response · Authors · 2021-08-31
> > > > **Thank you**
> > > >
> > > > Thank you for your response and reconsideration.
> > > >
> > > > We appreciate the time invested for the review and your suggestions.
> > > >
> > > > Re: computational benefits, we understand your standpoint and we could include a lengthier discussion on the topic in our paper in the appendix.

---

> ### Author Response · Authors · 2021-08-21
> **We would like to hear what you think about our response to your concerns.**
>
> Given the upcoming openreview deadline, we were kindly wondering whether our response to the reviewer and the extra experiments have addressed their concerns, and we would greatly appreciate a reply.

---

### Official Review · Reviewer_Pjs4 · 2021-07-16

**Rating:** 7
**Confidence:** 4

**Summary:**

The paper considers system heterogeneity in federated learning where different devices may have distinct processing capabilities or network bandwidth. Prior work addressed this scenario by not considering lower-end devices or scaling down models to fit all participants. In contrast, this paper proposes a novel (ordered) dropout technique to dynamically prune a shared model without sacrificing accuracy. Unlike random dropout, ordered dropout induces hierarchical pruning, allowing for efficient training via knowledge distillation. The paper empirically validates the proposed method on various federated learning benchmarks, demonstrating superior performance over the state-of-the-art.

**Ethical Concerns:**

There are no ethical issues with this paper.

**Limitations And Societal Impact:**

The paper fails to discuss potential negative societal impacts within the page limit (as required by the [NeurIPS blog](https://neuripsconf.medium.com/introducing-the-neurips-2021-paper-checklist-3220d6df500b)).

**Main Review:**

**Originality:** The paper proposes the first method addressing system heterogeneity in federated learning. To that end, the paper develops a novel form of dropout, which leverages a hierarchical structure to ensure good performance across a wide range of device capabilities. Related work is adequately cited.

**Quality:** The submission is technically sound, and the method is empirically validated on various federated learning benchmarks. However, the code was not submitted.

**Clarity:** The paper is clearly written and well organized. However, the section heading margins have been substantially decreased. I am not sure whether this is grounds for rejection (according to the [NeurIPS 2021 Style Files](https://media.neurips.cc/Conferences/NeurIPS2021/Styles/neurips_2021.pdf)), but it significantly reduces readability and should be changed.

**Significance:** Unlike prior work, which does not consider system heterogeneity or resorts to suboptimal approaches such as dropping lower-end devices or downscaling the shared model, the proposed method embraces heterogeneity. Consequently, the proposed technique is simple yet elegant and provides a practical solution to a relevant problem in federated learning.

**Questions**

- How do ordered dropout and knowledge distillation ordered dropout compare in terms of running time? Knowledge distillation might be significantly more expensive since it always evaluates the largest possible model.
- How do you achieve good accuracy with $\alpha = 1$ where the loss consists entirely of the KL divergence term?
- What do $U$ and $V$ correspond to in Theorem 1? After reading the proof, I assume that they denote the network layers, but the reader should not have to guess.
- How do you evaluate federated dropout or extended federated dropout on devices with low capabilities? While random dropout prunes neurons during training, these neurons are all present at inference time.

**Time Spent Reviewing:**

4

---

> ### Author Response · Authors · 2021-08-10
> **Response to the review**
>
> We would like to thank the reviewer for the constructive feedback and valuable time spent reviewing our manuscript. We are glad that the reviewer acknowledges the significance of the device heterogeneity challenges in FL and the execution of FjORD.
>
> **wrt FjORD vs FjORD w/ KD.** We acknowledge that the use of knowledge distillation (KD) does introduce a non-negligible computational overhead compared to pure ordered dropout (OD), nonetheless, we believe that it is justified as we argue below.
> We provide the alternative of FjORD w/ KD mainly as a solution for cases where the client bottleneck is memory or network related, rather than computational in nature. In such cases, the extra f/w and b/w propagations are effectively contributing only to the computational latency (always bound to be $\leq 2\times p^i_{\max}$), but the memory footprint of the model and upstream updates remains largely the same. However, in cases where client devices are computationally bound in terms of training latency, we propose FjORD without KD or decreasing $p^i_{\max}$ to account for the overhead of KD. In the centralised setting of OD training (Sec. 3.5), as well as for inference, the impact of KD is less of an issue. Lastly, compared to the status-quo of performing multiple training runs for each distinct model capacity, FjORD, w/ or w/o KD, greatly reduces the computational and communication cost by training all the width-based nested models in a single training run while providing better performance (e.g., Figure 4).
>
> **wrt $\alpha=1$.** We agree that relying solely on KL divergence would not work as it would provide no supervised feedback. Instead, in both our single-node and federated scenarios (see lines 119-121), we perform two backward propagations, one for the maximum-width model (teacher) with true labels and one for the sampled width model (student) with KD loss ($\alpha=1$ -> KL with teacher’s pseudo labels). Since the teacher already performs forward propagation to generate the pseudo labels, we take advantage of this computation to also backprop in the $p^i_{\max}$ model. In our experiments, we found this to perform better than vanilla single subnetwork backprop with $\alpha < 1$.
>
> **wrt U and V.** We apologise for missing reference in the main text. The reviewer is indeed correct in their assumption that these represent the layers. We will update this in the revised manuscript.
>
> **wrt the Random Dropout/eFD Inference.** We agree with the reviewer that both FD and eFD would not offer computational, memory or downstream gains at inference if they sent the whole global model. However, an alternative scenario is for the random dropout to happen server-side and then deploy the submodel to devices for inference. Naively with sparse representation of the pruned (with FD/eFD) network, one would expect no real gains in memory or computation unless the representation of the network changed (i.e. no bitmasking). In our evaluation, we simply evaluate centrally on a global validation set to report avg. accuracy of the global (sub)models. System heterogeneity aside, this is a major advantage for using OD as a latency-accuracy trade-off mechanism, as it enables a "train-once-deploy-everywhere" paradigm for inference with computational, memory and network bandwidth benefits out of the box.

---

### Official Review · Reviewer_WKyC · 2021-07-18

**Rating:** 7
**Confidence:** 5

**Summary:**

This paper offers a dropout approach to account for different client resources. The idea is simply through pruning the network where clients fit smaller networks. The papers also offers an approach called ordered dropout where end layers are dropped rather than random neurons.

**Ethics Review Area:**

["I don’t know"]

**Main Review:**

The paper idea is interesting. Below are my main concerns:

1) Starting from the title the paper claims this approach induces fairness. How and why ? Having clients with less computational power train smaller models does not guarantee any fairness. Perhaps it might even hurt it. The authors needs to clearly define a measure of fairness and if not prove at least test how this measure is improved. Ex: is the distribution of accuracy across clients narrowed ?

2) The authors are encouraged to check convergence. This idea of ordered pruning can induce bias in gradients, especially if clusters are imbalanced ! Will the method still recover a meaningful solution ?

3) The idea of clustering is strange. One would expect clusters to be done based on P_{y|x} at each client or perhaps P_{y,x}. The authors instead cluster and aggregate within clients of similar capabilities. What about heterogeneity is the data distribution ? what if clients is same clients (computation wise) are very different in data distribution

4) The authors need to benchmark with more models and personalization approaches.


**Time Spent Reviewing:**

1

---

> ### Author Response · Authors · 2021-08-10
> **Response to the review.**
>
> We thank the reviewer for their review and the concerns aired in their feedback. However, we believe that the reviewer has misunderstood core concepts of our paper and thus expects a different problem setting. Thus, we reiterate the objective and contributions of our paper and address the comments one by one.
>
> ### Summary of our paper's contributions
>
> The main objective of FjORD is to allow devices of different compute/memory/network capabilities to still participate in federated training, with the objective of training a globally good model that encompasses knowledge from all clients.
> The way this is accomplished is through Ordered Dropout, a method of importance-based pruning that allows the training of nested submodels of different widths and their extraction without the need to retrain. We apply this technique in the setting of FL, so as to allocate models of variable footprint to devices such that the participating clients can train in due time.
>
> Specifically, in our work:
> - Ordered Dropout does ordered, importance-based, **width-pruning**, rather than dropping "end layers".
> - When aggregating updates in FjORD, this is **not** done per device cluster. Aggregation happens across all updates irrespective of their p value (i.e. across devices, see Eq. 2, lines 188 - 195).
> We define fairness as the capability of FjORD to allow clients of various compute/memory/network capabilities to contribute to federated training, by updating smaller nested submodels.
> - Our goal is to leverage local client data to train a **single global model**. This means that we are not doing model personalisation, albeit it is indeed an interesting avenue of research.
>
> We discuss these topics in greater detail below.
>
> **wrt Ordered Dropout**.
> Nested submodels in our work are the result of width-pruning rather than dropping "end layers", as mentioned in the reviewer’s paper summary.
>
> **wrt Fairness**.
> As mentioned in Section 2, the status quo in FL is to either 1) drop low-tier devices as being stragglers, or 2) to use a single model of reduced capacity that fits the slowest device. The first introduces biases by always dropping weaker devices that might dominate in certain demographics, while the second can result in reduced model performance. "Fair" in FjORD comes from the fact that all devices are able to participate and all client data are considered, independently of their capabilities. We do not state that all clients contribute equally at a neuron level. We can refer to this as "fairness in participation", rather than "fairness in contribution" if the reviewer deems it necessary. Last but not least, via ordered representation of knowledge in OD, all clients get to at least influence the most impactful features of the model.
>
> **wrt Convergence**.
> We already provide evidence in the Appendix (Figure 8 and 9) that our ordered dropout approach leads to smoother convergence and lower losses across the three (model, dataset) combinations when compared to the state-of-the-art. Furthermore, we also show in Figure 7, that FjORD adapts well to different cluster distributions. In particular, for "drop scale=1", where most of the devices can run the full model, we show that this does not degrade the performance of the smaller subnetworks.
>
> **wrt Clustering**.
> The purpose of clustering is to tackle device heterogeneity by allowing us to sample submodels that are trainable on these devices. Under no circumstances do we only aggregate across devices of similar capabilities. Aggregation and updates are happening globally across p-values (and across device clusters). Essentially, device clusters are just a method for associating groups of devices of similar capabilities to p values. Without clustering, we would need to model each device as a separate $p$ value, which would not scale (lines 142-160).
>
> **wrt Data heterogeneity**.
> While we do not directly target the problem of data heterogeneity, we note that: 1) we do test our approach with heterogeneity in the data distribution through non-IID datasets; 2) allowing all devices to participate means even data from low-tier devices (previously considered stragglers) is included in the training, reducing biases; 3) we show empirically and theoretically that our ordered dropout exhibits ordered feature importance and since submodels are essentially the same model with reduced width and we aggregate all widths, all tiers get to at least influence the most impactful features of the model.
>
> **wrt Personalisation**.
> The clustering of devices is by no means for the purpose of personalisation, but rather to group together devices of similar capabilities into tiers, so that we can decompose the global model in a set of nested submodels with different compute requirements that fit within the capabilities of each of these device tiers (see lines 142-160).
> Finally, while we do not focus on personalisation, and explicitly tackle device heterogeneity rather than data heterogeneity, we agree that this is a good venue to pursue future research.
>
> **wrt Models**.
> We believe that our datasets and models, covering both vision and language modelling, are representative and widely used across federated learning research (e.g., [10, 32, 40]). That said, if the reviewer has a suggestion that is feasible to train within the given time limit, we are willing to provide results.
>
> We hope that the above clarifications convince the reviewer that this is indeed a relevant challenge and motivate the reviewer to reconsider the provided review and rating for our manuscript.

---

> > ### Comment · Reviewer_WKyC · 2021-09-01
> > **Final Evaluation**
> >
> > The reviewer's points are convincing especially the notion of using fairness of participation rather than fairness in contribution. I still suggest the authors to further explore why additional participation can improve fairness in the sense of uniform performance across devices. That being said fairness in participation is still worth investigating. I therefore raise my score to 7

---

### Public Comment · ~Yufei_CUI2 · 2022-04-12
**Similar dropout techniques named 'Ordered Dropout' have been proposed by published papers.**

Hi Authors,

Thanks for the nice work exploring nested networks under the federated learning framework.

This is a kind notice that:

1. The "Ordered Dropout" for generating nested sub-networks has been proposed our previous paper [1] published online in July 2020.
2. Our follow-up work, proposing "Variational Nested Dropout" with variational Bayes treatment for the uncertainty-calibrated nested sub-networks [2], was published in June 2021.
3. The patent [3] for "Ordered Dropout" was published in February 2022.

The two papers were both published before the revision and final version deadlines of NeurIPS 2021. We suggest fair comparisons with these works should be presented in a revised version in the NeurIPS paper or the arxiv manuscript.

Regards.

References:

[1] Cui, Y., Liu, Z., Yao, W., Li, Q., Chan, A. B., Kuo, T. W., & Xue, C. J. Fully Nested Neural Network for Adaptive Compression and Quantization. In 29th International Joint Conference on Artificial Intelligence (IJCAI 2020) (pp. 2080-2087).

[2]  Cui, Y., Liu, Z., Li, Q., Chan, A. B., & Xue, C. J. (2021). Bayesian Nested Neural Networks for Uncertainty Calibration and Adaptive Compression. In Proceedings of the IEEE/CVF Conference on Computer Vision and Pattern Recognition  (CVPR 2021) (pp. 2392-2401).

[3] Kuo, Tei-Wei, et al. "Artificial neural network configuration and deployment." U.S. Patent Application No. 17/000,612.

---

> ### Public Comment · Authors · 2022-06-16
> **Paper positioning**
>
> Dear Yufei,
>
> Thanks for kindly bringing your paper to our attention. While we recognise your publication about "Ordered Dropout" in IJCAI, we were unaware of your publication until this very moment. In fact, your paper does not appear under search of the "nested dropout" or "ordered dropout" terms.
>
> We are glad that more researchers are looking into similar topics as we have, but we respectfully disagree with your indications. Precedence in terms of publication of results is not exactly as you described in your previous comment, as we ourselves have evidence of prior submission of this work along with a patent filing. We would consider our works as "simultaneous submissions", especially at the time of writing. As an indication of good faith, we would be willing to include your work in related work of the arxiv version, indicating it as such.
>
> Finally, please note that although we use the same name, “Ordered Dropout,” our works have quite a few differences.  The main similarity is a structured dropout of nested structures. In our work, we only consider width-based pruning. In contrast, you went beyond that in your work and proposed using the same technique for residual connections and quantisation levels for limited precision.
>
> Let us now summarise the differences specific to width-based pruning. We do not apply any reweighting and use the pruned network without modifications. Furthermore, we noted that you need a searching procedure that require access to the validation dataset for deployment. In contrast, our approach only switches the width that we deploy based on the local constraints of the given device. We note that your search tool might provide better performance than our approach. Still, we don't assume access to representative validation datasets due to FL privacy constraints in our setting. We also realize that you proposed to drop weights randomly, but we note this does not perform well when performing width-based pruning (Channel-Cifar100 in Figure 4 of paper [1]).
>
> To conclude, we believe that the above, combined with specific deployment requirements of FL and the fact that FN$^3$ does not natively account for on-device training on heterogeneous clients (we realise this is beyond the original scope of your work), make our approaches extremely hard to compare directly.
>
>
> Best regards,
> FjORD authors

---

### Decision · Program_Chairs · 2021-09-27

**Decision:**

Accept (Spotlight)

**Comment:**

This paper develops new techniques for tackling system heterogeneity (i.e., diversity in the processing capabilities and network bandwidth of clients) in federated learning. The paper highlights the practical issue of system heterogeneity in federated learning that is often ignored in many papers. All the reviewers agree that the paper has interesting ideas and solves an important problem, and are in favor of accepting the paper. Along with data heterogeneity, systematic study of system heterogeneity can potentially have significant practical value and can be of interest to a wider audience. I recommend acceptance of the paper. I suggest the authors address the concerns of the reviewers in the final revision.